# Poisoning Retrieval Corpora by Injecting Adversarial Passages

**Zexuan Zhong**[†*]**, Ziqing Huang**[‡*]**, Alexander Wettig**[†]**, Danqi Chen**[†]
[†]Princeton University    [‡]Tsinghua University
{zzhong,awettig,danqic}@cs.princeton.edu    hzq001227@gmail.com

## Abstract

Dense retrievers have achieved state-of-the-art performance in various information retrieval tasks, but to what extent can they be safely deployed in real-world applications? In this work, we propose a novel attack for dense retrieval systems in which a malicious user generates a small number of adversarial passages by perturbing discrete tokens to maximize similarity with a provided set of training queries. When these adversarial passages are inserted into a large retrieval corpus, we show that this attack is highly effective in fooling these systems to retrieve them for queries that were not seen by the attacker. More surprisingly, these adversarial passages can directly generalize to out-of-domain queries and corpora with a high success attack rate—for instance, we find that 50 generated passages optimized on Natural Questions can mislead >94% of questions posed in financial documents or online forums. We also benchmark and compare a range of state-of-the-art dense retrievers, both unsupervised and supervised. Although different systems exhibit varying levels of vulnerability, we show they can all be successfully attacked by injecting up to 500 passages, a small fraction compared to a retrieval corpus of millions of passages.[1]

## 1 Introduction

Dense retrievers have become increasingly popular (Karpukhin et al., 2020; Xiong et al., 2021; Izacard et al., 2022; Khattab and Zaharia, 2020), and outperform traditional lexical methods in a range of information retrieval tasks. However, recent evidence shows that they may still struggle with long-tail entities (Sciavolino et al., 2021) or out-of-domain generalization (Thakur et al., 2021), and it remains unclear to what extent we can safely deploy dense retrievers in real-world applications.

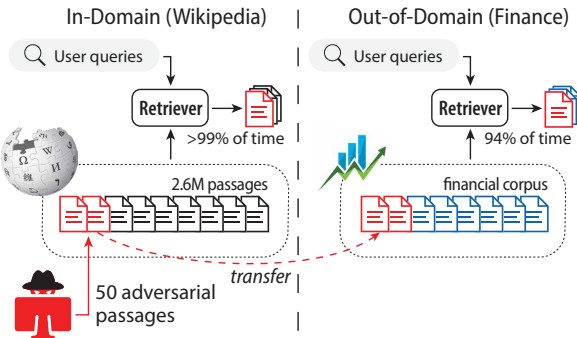

Figure 1: Our proposed *corpus poisoning* attack. Malicious users generate adversarial passages and inject them into a retrieval corpus to mislead dense retrievers to return them as responses to user queries. The attack is highly effective on unseen queries either in-domain or out-of-domain.

In this work, we reveal a new type of vulnerability in dense retrievers. We present a novel *corpus poisoning* attack, where a malicious user can inject a small fraction of adversarial passages to a retrieval corpus, with the aim of fooling systems into returning them among the top retrieved results. While previous work has shown that an adversarial passage can be crafted for individual queries (Song et al., 2020; Raval and Verma, 2020; Song et al., 2022; Liu et al., 2022), we consider a stronger setting, where an adversarial passage must be retrieved for a broad set of user queries, and can even generalize to unseen queries either in-domain or out-of-domain. This setting is realistic for open-access platforms, such as Wikipedia or Reddit, and such attacks may become a new tool for black hat SEO (Search Engine Optimization) (Patil Swati et al., 2013). Adversarial passages deteriorate the user experience of retrieval systems and can cause societal harm, e.g., if inserted passages contain misinformation.

Our attack is a gradient-based method, inspired by the HotFlip method (Ebrahimi et al., 2018), which starts from a natural-language passage and

---

[*]ZZ and ZH contributed equally. The work was done when ZH interned at Princeton University.

[1]Our code is publicly available at https://github.com/princeton-nlp/corpus-poisoning.

iteratively perturbs it in the discrete token space to maximize its similarity to a set of training queries. We also use a simple clustering method to extend this method to generate multiple passages.

We evaluate our method on state-of-the-art dense retrieval models and demonstrate that this corpus poisoning attack can be highly effective with a small number of adversarial passages. Noticeably, we find that unsupervised Contriever models are particularly vulnerable, and even 10 passages can fool more than $90\%$ of queries. Supervised retrievers (e.g., DPR, ANCE) seem to be harder to attack, but the success rate still exceeds 50% when we inject up to 500 passages (<0.02% compared to the retrieval corpus). Our attack is also effective at attacking multi-vector retrievers (e.g., ColBERT), which are less sensitive to single-token changes. Moreover, we find that these adversarial passages can be directly injected into out-of-domain retrieval corpora and achieve a high attack rate for unseen queries posed on these domains (e.g., financial documents or online forums), as illustrated in Figure 1.

We conclude our paper with a case study of real-world threats when the inserted text includes targeted misinformation. We hope that our findings will caution practitioners to think about the safety concerns of these dense retrieval systems, and further improve their robustness.

## 2 Related Work

In NLP, adversarial attacks have been studied on many classification tasks (Liang et al., 2018; Li et al., 2020; Morris et al., 2020; Wang et al., 2021) and question-answering tasks (Jia and Liang, 2017). The objective is to find small, semantic-preserving perturbations to the input that cause the model to make erroneous predictions.

Adversarial attacks on retrieval systems have been limited to making small edits to passages to change their retrieval rank for an individual or a small set of queries (Song et al., 2020; Raval and Verma, 2020; Song et al., 2022), and the success rate of the attack is evaluated per perturbed passage. In corpus poisoning, we inject new passages into the retrieval corpus and measure the attack success by the *overall performance* of the retrieval system when evaluated on *unseen* queries.

Corpus poisoning differs from data poisoning (Chen et al., 2017; Schuster et al., 2020), since changes are only made to the retrieval corpus rather than the training data, and the retriever remains unchanged. It should be noted that the attack of image retrieval systems has been explored in previous work (Xiao and Wang, 2021; Tolias et al., 2019), but those methods are not transferable to the text domain, where the optimization problem needs to be solved over discrete tokens, rather than continuous values of pixels.

Our work is related to universal adversarial triggers (Wallace et al., 2019; Song et al., 2021), which attack text classifiers by prepending the same phrase to any input sequence. However, in retrieval, we can change the system's output by injecting new passages, without having to alter every passage or user query, making it a more realistic setting for real-world attacks.

## 3 Method

**Problem definition**   We consider a passage retrieval problem: the retrieval model takes a user query $q$ and retrieves passages from a corpus $\mathcal{C}$, consisting of a set of passages $\{p_1, p_2, \ldots p_{|\mathcal{C}|}\}$. We mainly focus on dual-encoder dense retrievers that embed queries and passages using a query encoder $E_q(\cdot)$ and a passage encoder $E_p(\cdot)$ respectively, and compute their similarity using the inner product of the embeddings: $\mathrm{sim}(q, p) = E_q(q)^\top E_p(p)$. Additionally, in Section 4.4 we study our attack on multi-vector dense retrievers (e.g., ColBERT (Khattab and Zaharia, 2020)), where queries and passages are represented using multiple vectors.

The query and passage encoders are typically trained with a contrastive-learning objective, either from supervised question-passage pairs in the case of DPR (Karpukhin et al., 2020) and ANCE (Xiong et al., 2021), or from unsupervised data in the case of Contriever (Izacard et al., 2022). During inference, the dense retrieval models perform a nearest-neighbor search given a query $q$ to return the top-$k$ most similar passages in the corpus as retrieval results.

**Corpus poisoning attack**   In our corpus poisoning attack, a set of adversarial passages are generated and inserted into the retrieval corpus $\mathcal{C}$, with the intention of misleading dense retrieval models. Once the corpus has been poisoned, it is expected that dense retrieval models will retrieve the adversarial passages in response to certain queries. Specifically, we aim to generate a small set of adversarial passages $\mathcal{A} = \{a_1, a_2, \ldots, a_{|\mathcal{A}|}\}$, where the size of the adversarial passage set is much smaller than the original corpus, i.e., $|\mathcal{A}| \ll |\mathcal{C}|$. The objec-

tive is to construct $\mathcal{A}$ such that at least one adversarial passage is featured in the top-$k$ results for the distribution of queries $\mathcal{Q}$. In practice, we assume a training set from $\mathcal{Q}$ available to construct $\mathcal{A}$ and expect it to generalize to a held-out test set.

**Optimization** Given a set of queries $\mathcal{Q} = \{q_1, q_2, \ldots, q_{|\mathcal{Q}|}\}$, we aim to generate a set of adversarial passages $\mathcal{A}$ which are ranked high by the model in the retrieval results. To mislead the model, we would like to find a sequence of tokens $a = [t_1, t_2, \ldots]$ which maximizes the similarity to a set of queries:

$$a = \arg\max_{a'} \frac{1}{|\mathcal{Q}|} \sum_{q_i \in \mathcal{Q}} E_q(q_i)^\top E_p(a'). \quad (1)$$

We discuss how to generate a single passage first and extend it to generate multiple passages.

We use a gradient-based approach to solve the optimization problem, inspired by the HotFlip method (Ebrahimi et al., 2018; Wallace et al., 2019), which approximates the model output change of replacing a token. We initialize the adversarial passage using a random passage from the corpus. At each step, we randomly select a token $t_i$ in $a$ and compute an approximation of the model output if replacing $t_i$ with another token $t_i'$. We employ HotFlip to efficiently compute this approximation using gradients: $e_{t_i'}^\top \nabla_{e_{t_i}} \mathrm{sim}(q, a)$, where $\nabla_{e_{t_i}} \mathrm{sim}(q, a)$ is the gradient vector with respect to the token embedding $e_{t_i}$. Given a query set $\mathcal{Q}$, the best replacement candidates for the token $t_i$ can be obtained by selecting the token that maximizes the output approximation:

$$\arg\max_{t_i' \in \mathcal{V}} \frac{1}{|\mathcal{Q}|} \sum_{q \in \mathcal{Q}} e_{t_i'}^\top \nabla_{e_{t_i}} \mathrm{sim}(q, a), \quad (2)$$

where $\mathcal{V}$ is the vocabulary. Note that this operation is cheap as it only requires a single multiplication of the embedding matrix and the gradient vector.

**Generating multiple adversarial passages** We extend our attack to generate multiple adversarial passages. Our approach is based on a simple idea of grouping similar queries and generating one adversarial passage for each group of queries. To achieve this, we use the $k$-means clustering algorithm to cluster queries based on their embeddings $E_q(q_i)$. Once the queries are grouped into clusters, we generate one adversarial passage for each cluster by solving the optimization problem as shown

in Eq. 1. This allows us to generate multiple adversarial passages in parallel, each targeting a group of similar queries.

# 4 Experiments

## 4.1 Setup

**Retrieval datasets** We experiment with the BEIR benchmark (Thakur et al., 2021), a comprehensive suite of retrieval datasets. We focus on two popular datasets, Natural Questions (NQ) (Kwiatkowski et al., 2019) and MS MARCO (Nguyen et al., 2016), and evaluate our attack on the held-out test queries of these two datasets, as well as its transferability to seven unseen domains (e.g., Quora, scientific, financial documents) in BEIR. See Appendix A for details.

**Dense retrievers** In our main experiments, we conduct attacks on five state-of-the-art dense retrieval models: Contriever (Izacard et al., 2022) (pre-trained) and Contriever-ms (fine-tuned on MS MARCO), DPR-nq (Karpukhin et al., 2020) (trained on NQ) and DPR-mul (trained on multiple datasets), and ANCE (Xiong et al., 2021). Additionally, we apply the attack on a multi-vector dense retriever, ColBERT (Khattab and Zaharia, 2020), to study the effectiveness of our attack on different architectures.

**Evaluation metrics** We generate adversarial passages on the training set (NQ or MS MARCO) and inject them into the corpus and measure the top-$k$ *attack success rates* on test queries, defined as the percentage of queries for which at least one adversarial passage is retrieved in the top-$k$ results. A higher success rate indicates that a model is more vulnerable to corpus poisoning attacks. All the adversarial passages contain 50 tokens, and we perform the token replacement for 5000 steps. For more implementation details, see Appendix B.

## 4.2 Attacks on In-Domain Queries

Figure 2 (left) shows the in-domain attack results on different models in NQ and MS MARCO. We find that the pre-trained Contriever model (Izacard et al., 2022) is very vulnerable to our attack—even with only one adversarial passage, our attack fools the model on more than 75% of queries on NQ and MS MARCO. Although the supervised retrieval models (e.g., DPR-nq) appear harder to attack, the attack success rates can be substantially improved by generating more adversarial passages. In Figure 2 (right), we show that when we generate 500

|  | | NQ | | | MS MARCO | |
|---|---|---|---|---|---|---|
| $\|\mathcal{A}\| =$ | 1 | 10 | 50 | 1 | 10 | 50 |
| **Model** | | | | | | |
| Contriever | **84.2** | **98.1** | **99.4** | **75.2** | **92.2** | **98.6** |
| Contriever-ms | 0.5 | 52.5 | 80.9 | 2.4 | 20.9 | 34.9 |
| DPR-nq | 0.0 | 3.8 | 18.8 | 0.1 | 2.6 | 13.9 |
| DPR-mul | 0.0 | 10.6 | 28.3 | 0.0 | 4.7 | 16.3 |
| ANCE | 1.0 | 14.7 | 34.3 | 0.0 | 2.3 | 11.6 |

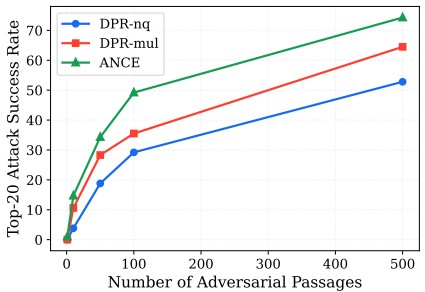

Figure 2: Top-20 success rate of our attack. Left: We generate $\|\mathcal{A}\| \in \{1, 10, 50\}$ adversarial passages on the train sets of NQ or MS MARCO and evaluate on their held-out test queries. Right: On NQ, we show that the attack success rate can be substantially improved by generating 500 passages. Contriever: the pre-trained Contriever model (Izacard et al., 2022), Contriever-ms: Contriever fine-tuned on MS MARCO, DPR-nq: DPR trained on NQ (Karpukhin et al., 2020), DPR-mul: DPR trained on a combination of datasets. ANCE: (Xiong et al., 2021).

| Source domain | | NQ | MS |
|---|---|---|---|
| **Target domain** | $\|\mathcal{C}\|$ | | |
| NQ | 2.6M | - | 98.7 |
| MS MARCO | 8.8M | 95.3 | - |
| Hotpot QA | 5.2M | **100.0** | **100.0** |
| FiQA | 57K | 94.1 | 96.8 |
| Quora | 522K | 97.2 | 98.3 |
| FEVER | 5.4M | 97.9 | 86.4 |
| TREC-COVID | 171K | 90.0 | **100.0** |
| ArguAna | 8.6K | 69.6 | 69.8 |
| SCIDOCS | 25K | 76.1 | 71.4 |

Table 1: Transferring top-20 success rate of our attack with $\|\mathcal{A}\| = 50$ adversarial passages. We conduct the attack on Contriever with NQ or MS MARCO (MS) training sets and evaluate the attack on different test sets of the BEIR benchmark (Thakur et al., 2021). See Appendix A for more details of each dataset.

adversarial passages, all dense retrieval models are fooled on at least $50\%$ of test queries.[2] Our results clearly demonstrate that dense retrieval models are significantly vulnerable to our proposed attack and fail on a large number of queries with only a small fraction of adversarial passages.

### 4.3 Attacks Transfer Out-of-Domain

We test whether the generated adversarial passages can transfer across different domains and retrieval tasks. We perform the attack on the Contriever model with the training sets of NQ and MS MARCO, and insert the adversarial passages into the corpora of other retrieval tasks. Table 1 shows the transfer attack results. Surprisingly, we find that the generated adversarial passages can transfer effectively to mislead the models on other retrieval domains/datasets (e.g., the attack success rate is $94.1\%$ on FiQA, a financial domain question-

---

[2]Due to the limited computing resources, we use $\|\mathcal{A}\| = 50$ in most of the experiments in the rest of paper.

|  | | NQ | |
|---|---|---|---|
| $\|\mathcal{A}\| =$ | 1 | 10 | 50 |
| ColBERT (50 tokens) | 0.1 | 3.0 | 11.5 |
| ColBERT (250 tokens) | 0.1 | 8.5 | 20.1 |

Table 2: Top-20 success rate of our attack on Col-BERT (Khattab and Zaharia, 2020). We generate adversarial passages with $\{50, 250\}$ tokens on the train set of NQ. We find the attack performance on multi-vector retrievers is more sensitive to adversarial passage lengths.

answering task). This implies a potential for practical deployment of the attack that is effective in different domains.

### 4.4 Attacks on Multi-Vector Retrievers

In Section 4.2, we conduct attacks on state-of-the-art dense retrieval models that map a query or passage to a single vector representation. Here, we study the effectiveness of our attack on multi-vector retrievers, where a piece of text corresponds to multiple vectors. We conduct our attack on Col-BERT (Khattab and Zaharia, 2020) (ColBERT-v1, pre-trained on MS MARCO) using the training queries from NQ with $\{1, 10, 50\}$ adversarial passages. Table 2 shows the attack performance on ColBERT on in-domain NQ queries. Interestingly, we find that generating longer adversarial passages (250 tokens) is much more effective in attacking ColBERT than using generating short ones (50 tokens). We think this is because the similarity function is computed with multiple embeddings in multi-vector dense retrieval models (e.g., SumMax in ColBERT), thus it is less sensitive to single token changes. We find that inserting 50 adversarial passages with 250 tokens fools ColBERT on $20.1\%$ NQ test queries, suggesting that our attack is still effective in attacking multi-vector retrieval models.

| Length (tokens) | 10 | 20 | 50 | 100 |
|---|---|---|---|---|
| Top-20 success rate | 63.5 | 81.6 | 80.9 | 79.2 |

Table 3: Top-20 success rate of our attack with Contriever-ms on the test queries of NQ when using different lengths of adversarial passages. We generate 50 adversarial passages using the train set of NQ.

### 4.5 Targeted Attack with Misinformation

Finally, we consider a case study, in which we examine the possibility of generating adversarial passages that include targeted misinformation. To accomplish this, we initialize the adversarial passages with "OUR OWN GOVERNMENT IS LITERALLY POISONING US WITH CHEMTRAILS" followed by [MASK] tokens and keep the prefix fixed during optimization. We experiment with the Contriever model using the NQ dataset by generating $|\mathcal{A}| \in \{1, 10\}$ of such adversarial passages. We find the attack remains effective under this constrained setup, although the success rates drop, reaching 11.7% and 59.6% with 1 and 10 passages, respectively. This suggests users would see the targeted message after just two or three queries, demonstrating the potential for spreading harmful information this way.

## 5 Additional Analysis and Discussion

**Attacks are not model-agnostic** We study whether the generated adversarial passages transfer across different models. We find limited transferability (i.e., success rates < 0.5%) across similar models (e.g., Contriever to Contriever-ms) and no transferability across different model families. These results suggest that it would be challenging to leverage transferability for black-box attacks, a common strategy against real-world systems with unknown model weights (Papernot et al., 2016).

**Lengths of adversarial passages** We study the effects of adversarial passage lengths on the attack performance. We attack Contriever-ms using the training data from NQ with 50 adversarial passages of length $\{10, 20, 50, 100\}$. As Table 3 shows, as long as the adversarial passages contain a sufficient number of tokens (i.e., $\geq 20$), the attack performance is not affected much when further increasing the passage length.

**Initialization of adversarial passages** In our attack, we initialize the adversarial passage using a random passage from the corpus. Here, we study the importance of initializing the adversarial passages using natural text. We initialize the adversarial passages with 50 [MASK] tokens, the attack success rate drops from 98.1% to 95.8% when attacking Contriever on NQ with 10 adversarial passages. This shows that the attack is still effective with a nonsense initial text but using natural passages would make the attack more effective.

**Unnaturalness of adversarial passages** Our attack results in unnatural passages (see examples in Appendix E). This can be leveraged to formulate simple defenses, as we show in Appendix C. However, we note that attackers may develop stronger attacks which produce fluent text that can still fool models, as demonstrated by Song et al. (2021).

**Attacks on sparse retrievers (BM25)** We study whether the proposed attack is effective at sparse lexical retrieval models such as BM25. Specifically, we attack BM25 by maximizing the BM25 similarity of the adversarial passage to a group of queries (using Contriever embeddings in $k$-means). We find that the adversarial passage to a query group contains only repeated tokens, such as a passage consisting of 50 tokens "what". We generate and insert 50 adversarial passages and find that BM25 retrieval performance does not change at all (i.e., attack success rate = 0%).

Next, we study whether the adversarial passages for Contriever can transfer to BM25. We find that on the NQ dataset with 50 adversarial passages, 0% of queries are fooled by the Contriever adversarial passages, when we use BM25 to retrieve. These results suggest that sparse retrieval models are robust to the proposed corpus poisoning attack.

**Transferability does not simply come from train-test overlap** In Section 4.3, we show that the adversarial passages can transfer across different domains/datasets. In Appendix D, we study different transfer attack strategies and show that the transferability does not simply come from the overlap between training queries and test queries.

## 6 Conclusions

We proposed a new attack for dense retrievers, in which adversarial passages are inserted into the corpus to mislead their retrieval outputs. We show that even a small number of adversarial passages can successfully attack state-of-the-art dense retrievers and generalize to queries from unseen domains. These findings have important implications for the future deployment of robust retrieval systems in real-world applications.

## Limitations

The limitations of our research are as follows:

- Despite the effectiveness, our approach can be computationally expensive to carry out. For each group of queries, we need to run our attack independently on a single GPU to generate one adversarial passage. For future work, it is worth considering the exploration of more efficient methods for generating multiple adversarial passages, which may serve to mitigate the computational costs of our approach.

- Our study focuses on corpus poisoning attacks to investigate whether dense retrieval models can be deployed safely. We acknowledge that there exist other potential avenues of attack against dense retrieval models, such as maliciously manipulating user queries. Our investigation specifically centers on corpus poisoning attacks and their implications for the vulnerability of dense retrieval models.

## Ethical Considerations

Our research studies the vulnerability of dense retrieval models, which are widely used techniques in the industry. The results of our experiments indicate that malicious users can successfully perform the proposed attack, resulting in a high rate of misleading a dense retrieval model. Although we propose potential defenses in Section 5, it is important to note that the proposed attack has the potential for misuse, allowing for the spread of toxic information. Future research based on this attack should exercise caution and consider the potential consequences of any proposed method. Furthermore, we point out that we include a small piece of misinformation in the paper (the prefix in Section 4.5) to highlight the harmful potential of corpus poisoning.

## Acknowledgements

We thank Dan Friedman, Victoria Graf, and Yangsibo Huang for providing helpful feedback. This research is supported by the "Dynabench Data Collection and Benchmarking Platform" award from Meta AI. ZZ is supported by a JP Morgan Ph.D. Fellowship.

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

## A  Dataset Details

We use several datasets with different tasks and domains from BEIR (Thakur et al., 2021), the widely-used benchmark for zero-shot evaluation of information retrieval, to evaluate the transferability of the adversarial passages: Natrual Questions (Kwiatkowski et al., 2019), MS MARCO (Nguyen et al., 2016), HotpotQA (Yang et al., 2018), FiQA (Maia et al., 2018), Quora, FEVER (Thorne et al., 2018), TREC-COVID (Voorhees et al., 2021), ArguAna (Wachsmuth et al., 2018), and SCIDOCS (Cohan et al., 2020). We include more information and data statistics in Table 4.

## B  Implementation Details

We implement our attack based on the PyTorch library (Paszke et al., 2019). We set the length of all adversarial passages to be 50. During optimization, we perform token replacement for $5,000$ steps. At each step, we compute the gradients based on a batch of queries $\mathcal{Q}$ (we set batch size as 64) and randomly select a token $t_i$ in the current adversarial passage to be replaced. We use the gradient-base HotFlip method (Ebrahimi et al., 2018) to obtain a set of replacement candidates, which maximize the approximate similarity after replacement:

$$\operatorname*{arg\,max}_{t_i' \in \mathcal{V}} \frac{1}{|\mathcal{Q}|} \sum_{q \in \mathcal{Q}} e_{t_i'}^\top \nabla_{e_{t_i}} \operatorname{sim}(q, a). \quad (3)$$

We consider top-100 tokens as potential replacements. We measure the actual similarity changes on each of these candidates and use the token, which maximizes the similarity between the passage and this batch of queries, to replace $t_i$.

## C  Defenses

We notice that the adversarial passages generated by our attack have two distinct characteristics: (1) they contain highly unnatural sequences, and this can be detected by likelihood scoring of an off-the-shelf language model, and (2) their embeddings have very high $\ell_2$-norms to boost their similarity scores. We study potential approaches to detect adversarial passages.

**Filter by likelihood**  Adversarial passages contain highly unnatural sequences, see examples in Appendix E. One could easily detect and distinguish the adversarial examples from the corpus passages by scoring the average token log likelihood of each passage with a language model, not

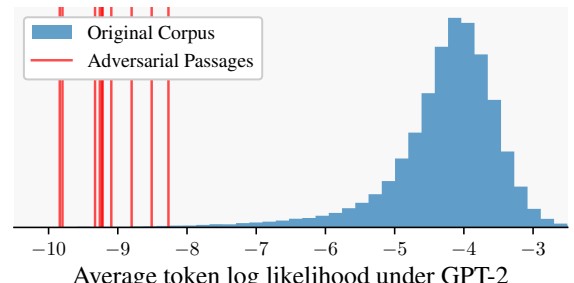

Figure 3: Average token log likelihood for Wikipedia passages and 10 corresponding adversarial passages for NQ with Contriever.

to mention spam filters and moderators on real-world platforms. Indeed, Figure 3 shows that an off-the-shelf GPT-2 can separate the corpus passages and adversarial passages almost perfectly. However, future work might show that an adversary can overcome this weakness by incorporating natural language priors into the attack.

**Embedding norm clipping**  The interaction between queries and passages is strictly limited to the inner product, such that $\operatorname{sim}(p, q) \propto \|E_p(p)\|_2 \cos\theta$, where $\theta$ is the angle between query and passage embedding. As it is not possible to reduce the angle with respect to a wide set of queries to zero, the attacker should primarily focus on producing passage embeddings with large $\ell_2$-norms. Indeed, Figure 4 shows that this is the case for the adversarial passages against Contriever. This prompts a second defense, where we clip the embedding norms of the passage at a constant $\alpha$ for all passages in the retrieval corpus:

$$E_{clip}(p) = E_p(p) \frac{\|E_p(p)\|_2}{\min(\|E_p(p)\|_2, \alpha)}. \quad (4)$$

Based on the scale observed as in Figure 4, we explore different values of $\alpha$ in Table 5 (in Appendix C) and find that clipping with $\alpha = 1.75$ reduces the attack success rate by $99.4\%$ while only hurting the standard retrieval performance by $6\%$. Despite this promising defense, it remains possible that corpus poisoning for targeted query distributions or more sophisticated architectures, e.g,. ColBERT's late interaction mechanism (Khattab and Zaharia, 2020), may be harder to detect via vector norm anomalies.

| Dataset | Task | Domain | # Query | # Corpus |
|---|---|---|---|---|
| Natural Questions (NQ) (Kwiatkowski et al., 2019) | Question Answering (QA) | Wikipedia | 132,803 (train) 3,452 (test) | 2,681,468 |
| MS MARCO (MS) (Nguyen et al., 2016) | Passage Retrieval | Misc. | 532,761 (train) 6,980 (test) | 8,841,823 |
| HotpotQA (Yang et al., 2018) | QA | Wikipedia | 170,000 (train) 7,405 (test) | 5,233,329 |
| FiQA (Maia et al., 2018) | QA | Finance | 648 | 57,638 |
| Quora | Question Retrieval | Quora | 10,000 | 522,391 |
| FEVER (Thorne et al., 2018) | Fact Checking | Wikipedia | 140,085 (train) 6,666 (test) | 5,416,568 |
| TREC-COVID (Voorhees et al., 2021) | Information Retrieval | Bio-Medical | 50 | 171,332 |
| ArguAna (Wachsmuth et al., 2018) | Argument Retrieval | Misc. | 1,406 | 8,674 |
| SCIDOCS (Cohan et al., 2020) | Citation Prediction | Scientific | 1,000 | 25,657 |

Table 4: Statistics of datasets. Most of the information comes from BEIR (Thakur et al., 2021).

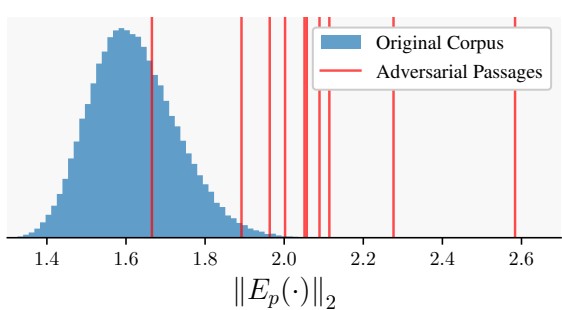

Figure 4: Distribution of $\ell_2$-norms for embeddings of Wikipedia passages and 10 corresponding adversarial passages for NQ with Contriever.

| $\alpha$ | Retrieval acc. w/o attack | Attack success rate |
|---|---|---|
| $\rightarrow \infty$ | 56.7 | 99.4 |
| 2.25 | 52.4 | 7.2 |
| 2.00 | 52.6 | 3.2 |
| 1.75 | 53.4 | 0.6 |
| 1.50 | 52.2 | 0.2 |
| $\rightarrow 0$ | 49.8 | 0.0 |

Table 5: Top-20 retrieval accuracy (w/o attack) and attack success rate when clipping the $\ell_2$-norms of passage embeddings at a value $\alpha$ for a Contriever model on NQ. $\alpha \rightarrow \infty$ denotes no clipping and $\alpha \rightarrow 0$ is equivalent to using cosine similarity.

| | Full transfer | (1) | (2) |
|---|---|---|---|
| **NQ (in-domain)** | 99.4 | 93.9 | 94.2 |
| **Out-of-domain** | | | |
| MS MARCO | 95.3 | 77.1 | 76.3 |
| Hotpot QA | 100.0 | 97.3 | 97.9 |
| FiQA | 94.1 | 74.5 | 74.0 |
| Quora | 97.2 | 81.0 | 80.2 |
| FEVER | 97.9 | 63.9 | 82.3 |
| TREC-COVID | 90.0 | 68.0 | 72.0 |
| ArguAna | 69.6 | 23.5 | 22.2 |
| SCIDOCS | 76.1 | 25.2 | 28.6 |

Table 6: Performance of different transfer attack strategies. **Full transfer**: the vanilla transfer attack we used in Section 4.3. **(1)**: transfer attack via the most similar training query. **(2)**: transfer attack via the $k$-means cluster. See Appendix D for details.

# D   Transferability Does Not Simply Come from Train-Test Overlap

In Section 4.3, we show that the generated adversarial passages can transfer across different domains/datasets. Here we investigate whether the transferability simply comes from the overlap between training and testing query distributions.

We attack Contriever on the training set of NQ and generate 50 adversarial passages. For a given test query $q$ (in-domain or out-of-domain), we first find similar queries from the training set, and we use the adversarial passages that are most effective on those training queries to attack the current test query $q$. Specifically, we consider two cases: **(1)** We find the training query $q^*$, which is most similar to test query $q$ (measured using the query encoder of Contriever). Among 50 adversarial passages, we find the passage $p^*$ which maximizes the similarity to the training query $q^*$. We insert $p^*$ into the corpus as an attack. **(2)** We classify the test

query $q$ using the $k$-means classifier that is learned when generating adversarial passages. As for each $k$-means cluster, we have generated one passage, we simply insert the adversarial passage $p^*$ that is corresponding to the $k$-means cluster that $q$ is classified as.

The results are shown in Table 6. merely using (1) or (2) is not sufficient to preserve the transfer attack rates. For example, the performance on attacking Scidocs drops from $76.1\%$ to $25.2\%$, showing that the test queries and training queries are likely drawn from very different domains. We also find that the performance on HotpotQA does not drop a lot, because the distribution is similar to the training set (both NQ and HotpotQA include knowledge-related questions on Wikipedia). These results suggest that the high transfer attack success rates achieved by our approach do not simply come from the overlap between test queries and training queries.

## E   Qualitative Examples of Adversarial Passages

Table 7 shows some qualitative examples of adversarial passages generated by conducting the attack on Contriever (Izacard et al., 2022) with a group of NQ queries clustered by $k$-means. We find that although the generated passages are difficult to interpret as human text, they contain some keywords (highlighted in blue) that are highly relevant to the queries. For example, in a group of queries related to basketball, the corresponding adversarial passage includes words like "bulls" (a famous basketball team in the US) and "russell" (a famous basketball player).

| Clustered Query Examples | Adversarial Passage |
|---|---|
| what is the definition of an nba rookie
who has more nba championships the east or west
how many points did lebron have in the finals
when did dell curry retire from the nba | oblast exceed continuity improvement scenes cinematography enhance draftity rich dionual 750 harta pour bracelet rookie russell arbitration outrageous bull mega litter bulls elk parades scanning outta kr pound posts ; pronounced ive groove except how tap elite baskets |
| who played guitar in while my guitar gently weeps
who wrote the song all i have to offer you is me
who played guitar on you really got me
who wrote the song you're the one that i want | drug council deadlineanalysis composition disclosure ability deadlineponexidelity virussh 2018 markogusmpt vampire tornadolusion children hindwings hurricane premiere county combolife patriarch thru robb rat murders youngest hedge fbi kristintzedel dies suspects overdose [ textiles 23 |
| how many finger lakes are there in new york state
how many local government we have in ogun state
how many judges are there in supreme court now
how strong was the 2011 earthquake in japan | parade peat etc etc maxi stra ul flies joo microwave directions ceilings spa good view view more please wi towels poo sunshine sg ient lu slight refrigerator zipper super airfields observation numbers waterfront gard |
| what is dutch elm disease and how does it spread
where in the plant does the calvin cycle take place
what is the purpose of a shunt in the brain
why we use bsa as standard for protein estimation | mini comment combe qui attract hosh presbyterian heart boot mma pw led cell el what synth ctric ida pw integrate sound dyed thanked visa ael platt director cell nervous growing rdial wley answer den chain cular coiled thy wing nationalists perceived vati nant textbook attention ral warnings involving |
| what is the richest woman in the us
what is the birth and death rate in america
what is the population of las cruces new mexico
what is the third most viewed video on youtube | ? ranking is lowest lica gible what est amount iest internationally quieter automobiles necessary francoise lake draft bonuses mainly tyne bonuses defeats bonuses > day summary sourced sourced card autobiography suns training malaga bonuses qualifying through player alaska what whose locality surrounding ately receives liest logram whether |
| who played the judge in the good place
who directed one flew over the cuckoos nest
who played the waitress in the movie michael
who played in fast and furious tokyo drift | female vate vegetables amino sin tama / = wikipedia fuss bian cent magazine ti cco din naire du fill uts brush aster bu oe whose whose remove black vegetables prefer spices main article ris communes guides ... kos actress translation , daryl actor who ~ took potassium tests |

Table 7: Qualitative examples of generated adversarial passages. We show the query examples in a group clustered by $k$-means from the NQ dataset (left) and the corresponding adversarial passage (right). We highlight the keywords in the adversarial passages that are highly connected to the queries.