# OpenReview forum: "Poisoning Retrieval Corpora by Injecting Adversarial Passages"
_EMNLP/2023/Conference — EMNLP 2023 Main_

### Official Review · Reviewer_eoXC · 2023-07-30

**Soundness:** 4

**Excitement:**

3: Ambivalent: It has merits (e.g., it reports state-of-the-art results, the idea is nice), but there are key weaknesses (e.g., it describes incremental work), and it can significantly benefit from another round of revision. However, I won't object to accepting it if my co-reviewers champion it.

**Missing References:**

Include more dense retrieval architectures listed in the reasons to reject

**Paper Topic And Main Contributions:**

This paper proposes an approach for generating documents to add to a collection that are likely to be retrieved by a dense retrieval bi-encoder model that represents a document with a single vector. This is done by adding a set of adversarial passages where the objective is to create the passages in such a way that 1 of a small set of adversarial passage will be ranked in the top k results for the distribution of queries. To find representative queries, the training queries are clustered and by starting with a random passage in the collection and at each step randomly replacing a token with another token so that the similarity between the represented query and document increases. Such adversarial passages frequently appear in the ranked lists of queries in the test set, both when evaluating on the test set of for the collection that was used to train the model and on other evaluation data set that are different from the training data.

**Reasons To Accept:**

A major strength of the paper is investigation of bi-encoder retrieval models susceptibility to poisoning by introducing artificial documents that are frequently retrieved. The process of creating a passage and inserting it into a collection represents a vector of attack on such models if they were deployed by search engines. The paper clearly and succinctly describes the technique in a way that another researcher to the requisite computing skills should be able to replicate the work. In addition the paper identifies a particular weakness in this type of model. While the generated passages are non-sensical, they are highly ranked due to L2 regularization.

**Reasons To Reject:**

A weaknesses of this paper is that apart for a brief exploration of GPT3, the authors only include one type of architecture for dense retrieval, the bi-encoder architecture where passages are represented with a single dense vector. The paper needs to be clear that the paper cannot say anything about cross-encoder architectures, learned sparse representations (e.g SPLADE), or even bi-encoder architectures that represent queries and passages with multiple dense vectors, such as ColBERT. The abstract and introduction both need to be clear that the findings only apply to this single type of architecture for dense retrieval. The paper needs to introduce other dense architectures and cite the appropriate literature. Experiments with ColBERT need to be added to the Appendix.

A second weakness of the paper arises in the evaluation on the BIER dataset. The setup of the evaluation appears valid; however, the authors fail to investigate the similarity of queries in the different collections relative to the training data. The fact that documents are vastly different from the training set is immaterial if the queries are represented in the training data. In such a case, it would be unsurprising that the artificial passages are ranked highly as is seen in the evaluation. Such an investigation should be added to the paper to give the reader a better understanding of the poisoning vulnerability.

**Reproducibility:**

4: Could mostly reproduce the results, but there may be some variation because of sample variance or minor variations in their interpretation of the protocol or method.

**Reviewer Confidence:**

5: Positive that my evaluation is correct. I read the paper very carefully and I am very familiar with related work.

**Typos Grammar Style And Presentation Improvements:**

Remove the footnote from the abstract. The abstract needs to stand along, so it should include the footnoted information in the body of the abstract.

---

> ### Author Rebuttal · Authors · 2023-08-29
>
> We appreciate the invaluable comments from the reviewer. We are encouraged to see that the reviewer views our investigation of attacking retrieval models as a major strength, and the reviewer finds our paper describes the technique clearly and succinctly, and has studied potential defenses to this type of attack! We have addressed the comments in a point-by-point manner below. We are more than willing to engage in further discussions with the reviewers should any follow-up questions arise.
>
> > **“The abstract and introduction both need to be clear that the findings only apply to this single type of architecture for dense retrieval”**
>
> Thanks for the suggestion! We will revise the abstract and introduction to make our claims more precise that the majority of the work is to focus on bi-encoder architectures of dense retrieval models. However, we would like to note the importance and wide application of dual-encoder retrievers due to their simplicity and efficiency, and we have explored a range of popular models in this category.
>
> On the other hand, we thank the reviewer’s suggestion and added additional experiments based on ColBERT. To study the effectiveness of our attack on ColBERT, we conduct our attack on the ColBERT model (ColBERT v1, trained on MS MARCO) using the training queries from NQ with {1, 10, 50} adversarial passages. The following table shows the top-20 success rates of our attack on NQ test queries. We find that our attack is still effective in attacking multi-vector retrieval models like ColBERT. Interestingly, we find that generating longer adversarial passages (e.g., 250 tokens) is more effective in attacking ColBERT than using short passages (e.g., 50 tokens), likely because of the SumMax operation that ColBERT uses, while the performance of attacking single-vector models is not sensitive to adversarial passage lengths. We are very happy to include this experiment and discussion in our revision.
>
> | $\|\mathcal{A}\|$       | 1    | 10   | 50   |
> |-------------------------|------|------|------|
> | Contriever              | 84.2 | 98.1 | 99.4 |
> | Contriever-ms           | 0.5  | 52.5 | 80.9 |
> | DPR-nq                  | 0.0  | 3.8  | 18.8 |
> | DPR-mul                 | 0.0  | 10.6 | 28.3 |
> | ANCE                    | 1.0  | 14.7 | 34.3 |
> | ColBERT (250 tokens)    | 0.1  | 8.5  | 20.1 |
> |   - ablation: 50 tokens | 0.1  | 3.0  | 11.5 |
>
> **About attacking sparse retrieval models**: We note that in Appendix C.1, we have shown that the proposed attack is not effective in attacking BM25. The sparse retrieval model SPLADE expanded the sparse vector representations by leveraging the output logit vectors returned by BERT. We agree that it is worth studying the corpus poisoning attack to SPLADE, and we hope to do it in the future.
>
> >**“The authors fail to investigate the similarity of queries in the different collections relative to the training data.”**
>
> We appreciate the reviewer’s suggestion to investigate the similarity of test queries to the training queries. We have conducted additional experiments to investigate this issue.
>
> Suppose that we generate 50 adversarial passages to Contriever on the training set of NQ; for a given test query $q$, we first find similar queries from the training set, and we use the adversarial passages that are most effective on those training queries to attack the current test query $q$. Specifically, we consider two cases:
>
> **(1)** We find the training query $q^*$, which is most similar to test query $q$ (measured using the query encoder of Contriever). Among 50 adversarial passages, we find the passage $p^*$, which maximizes the similarity to the training query $q^*$. We insert $p^*$ into the corpus as an attack.
>
> **(2)** We classify the test query $q$ using the k-means classifier that is learned when generating adversarial passages. As for each k-means cluster, we have generated one passage; we simply insert the adversarial passage $p^*$ that corresponds to the k-means cluster that $q$ is classified as.
>
> Our hypothesis is that if the high transfer attack success rates mainly come from the overlap between test query distribution and train query distribution, the attack in (1) and (2) should be as effective as the full transfer attack. However, as the table below shows, merely using (1) or (2) is not sufficient to preserve the transfer attack rates. For example, the performance on attacking Scidocs drops from 76.1% to 25.2%, showing that the test queries and training queries are likely drawn from very different domains. We also find that the performance on HotpotQA does not drop a lot because the distribution is similar to the training set (both NQ and HotpotQA include knowledge-related questions on Wikipedia).
>
> These results suggest that the high transfer attack success rates achieved by our approach do not simply come from the overlap between test queries and training queries.  We hope these results provide a better understanding of the generalization of the adversarial passages. We are more than willing to engage in further discussions with the reviewers should any follow-up questions arise!
>
> |                   | Full transfer attack | **(1)** attack via closest training query $q^*$ | **(2)** attack via the k-means cluster |
> |-------------------|--------------------------|-----------------------------------------------------------|----------------------------------------|
> | NQ (**in-domain**)    | 99.4                     | 93.9                                                      | 94.2                                   |
> | **Out-of-domain** |                          |                                                           |                                        |
> | MS MARCO          | 95.3                     | 77.1                                                      | 76.3                                   |
> | HotpotQA          | 100.0                    | 97.3                                                      | 97.9                                   |
> | FiQA              | 94.1                     | 74.5                                                      | 74.0                                   |
> | Quora             | 97.2                     | 81.0                                                      | 80.2                                   |
> | FEVER             | 97.9                     | 63.9                                                      | 82.3                                   |
> | TREC-COVID        | 90.0                     | 68.0                                                      | 72.0                                   |
> | ArguAna           | 69.6                     | 23.5                                                      | 22.2                                   |
> | Scidocs           | 76.1                     | 25.2                                                      | 28.6                                   |

---

### Official Review · Reviewer_CaDC · 2023-08-05

**Soundness:** 3

**Excitement:**

3: Ambivalent: It has merits (e.g., it reports state-of-the-art results, the idea is nice), but there are key weaknesses (e.g., it describes incremental work), and it can significantly benefit from another round of revision. However, I won't object to accepting it if my co-reviewers champion it.

**Missing References:**

It needs to add more STOA works

**Paper Topic And Main Contributions:**

The paper presents an adversarial attack method for dense retrievers for IR system. The approach adopts the idea from HotFlip and successfully injects perturbed passages to fool the retriever.


**Questions For The Authors:**

Pls see "Reasons To Reject"

**Reasons To Accept:**

1. Effective attack method
2. Extensive empirical studies
3. Paper is well- presented and easy to follow.

**Reasons To Reject:**

1. The attack method is similar to HotFlip. So what is the technical contribution here? (except for different applications)
2. It is good to know the attack rate of different lengths of the injected passages.
3. Lack of comparisons to STOA works.

**Reproducibility:**

5: Could easily reproduce the results.

**Reviewer Confidence:**

4: Quite sure. I tried to check the important points carefully. It's unlikely, though conceivable, that I missed something that should affect my ratings.

---

> ### Author Rebuttal · Authors · 2023-08-29
>
> We thank the reviewer for the valuable comments. We are glad to see that the reviewer finds our paper is well-presented and has introduced an effective attack method and extensive empirical studies! We addressed the reviewer’s comments below and we are willing to engage in further discussion should any follow-up questions arise!
>
>
> >**“The attack method is similar to HotFlip. So what is the technical contribution here? (except for different applications)”**
>
> We do not claim that we propose a new method that can attack a wide range of NLP models (like HotFlip does); instead, we present a novel attack paradigm (i.e., corpus poisoning attack) for retrieval models. Compared to previous works which generate an adversarial passage for a specific query (e.g., Song et al., 2020, Song et al., 2022), we consider a stronger setting where the adversarial passage needs to fool the model on a broad set of user queries. Furthermore, our experimental results reveal the significant vulnerability of dense retrieval models and hope our research can facilitate the development of more robust dense retrieval models.
>
> >**“It is good to know the attack rate of different lengths of the injected passages.”**
>
> Our preliminary experiments showed that the attack performance was not sensitive to the length of adversarial passage.
>
> To address the reviewer’s concern, we additionally conduct an ablation study, where we attack Contriever-ms using the training data from NQ with 50 adversarial passages of length {10, 20, 50, 100}. As the following table shows, as long as the adversarial passages contain a sufficient amount of tokens (i.e., >=20), the attack performance won’t change much when further increasing the passage length. We will include this ablation study in our revision.
>
> | **Length**             | 10   | 20   | 50   | 100  |
> |------------------------|------|------|------|------|
> | **Attack performance** | 63.5 | 81.6 | 80.9 | 79.2 |
>
> >**“lack of comparisons to SOTA works.”**
>
> We are not sure which works the reviewer is referring to. We kindly request the reviewer to elaborate on which works we should compare with and it will significantly help us address the reviewer’s concern.
>
> Note that we present a novel attack paradigm where an adversarial passage must be retrieved for a set of user queries. Our setting is different from previous work (e.g., Song et al., 2020, Song et al., 2022), where an adversarial passage is crafted for a specific query.
>
> **References**
>
> Congzheng Song, Alexander Rush, and Vitaly Shmatikov. “Adversarial semantic collisions.” EMNLP 2020
>
> Junshuai Song, Jiangshan Zhang, Jifeng Zhu, Mengyun 481 Tang, and Yong Yang. “TRAttack: Text rewrit ing attack against text retrieval.” Proceedings of 483 the 7th Workshop on Representation Learning for NLP, 2022.

---

### Official Review · Reviewer_oe2c · 2023-08-07

**Soundness:** 4

**Excitement:**

4: Strong: This paper deepens the understanding of some phenomenon or lowers the barriers to an existing research direction.

**Paper Topic And Main Contributions:**

This paper present a novel attack for dense retrieval systems, in which they are able to effectively inject poisoned passages into the top-k documents retrieved by the systems. This method requires only a very small subset of the total training set to be poisoned to be effective.

**Reasons To Accept:**

- Well written, clearly defining their objective function and detailing how the method works
- Tests on a multiple QA datasets and two different retrieval models
- Used the BEIR benchmark to test transferring attacks
- For a short paper, provides experimentation to prove the effectiveness of the method and even provides basic experiments for defense

**Reasons To Reject:**

- None as a short paper

**Reproducibility:**

4: Could mostly reproduce the results, but there may be some variation because of sample variance or minor variations in their interpretation of the protocol or method.

**Reviewer Confidence:**

4: Quite sure. I tried to check the important points carefully. It's unlikely, though conceivable, that I missed something that should affect my ratings.

---

> ### Author Rebuttal · Authors · 2023-08-29
>
> We thank the reviewers for the valuable feedback! We are encouraged that the reviewer found our paper was well-written, with a clear objective and detailed method, and sufficient experiments to show the effectiveness of our approach!

---

### Meta-Review · Area_Chair_to7Q · 2023-09-19

**Recommendation:** 3

**Metareview:**

The paper presents an attack on dense retrieval systems, introducing a method to inject poisoned passages into the top-k documents retrieved by these systems. The attack method is thoroughly explained and extensively tested on multiple QA datasets and two different retrieval models, with additional evaluation using the BEIR benchmark for transferring attacks. The paper is well-written and structured, offering clear explanations of the method and its results, making it easy to follow.

One of the paper's significant strengths is its clear presentation of the attack method, its objectives, and the detailed explanation of how the method operates. The extensive empirical studies conducted by the authors provide strong evidence of the attack's effectiveness, enhancing the paper's overall quality. Additionally, the paper goes beyond the attack itself by including basic experiments for defense strategies, demonstrating a comprehensive approach to addressing the issue.

However, some concerns were raised by reviewers. One reviewer pointed out similarities between the attack method and the existing HotFlip technique, seeking clarification regarding the technical contributions of the paper. Additionally, the paper lacks comparisons to state-of-the-art (STOA) works in the field. The paper's focus on a specific type of dense retrieval architecture, the bi-encoder model, should be made clear in the abstract and introduction, and the paper should acknowledge other relevant architectures and cite appropriate literature. Lastly, the evaluation on the BEIR dataset should consider the similarity of queries in different collections relative to the training data to provide a more accurate assessment of poisoning vulnerability.

---

### Decision · Program_Chairs · 2023-10-07

**Decision:**

Accept-Main

**Comment:**

The paper presents an attack on dense retrieval systems, introducing a method to inject poisoned passages into the top-k documents retrieved by these systems. The attack method is thoroughly explained and extensively tested on multiple QA datasets and two different retrieval models, with additional evaluation using the BEIR benchmark for transferring attacks. The paper is well-written and structured, offering clear explanations of the method and its results, making it easy to follow.

One of the paper's significant strengths is its clear presentation of the attack method, its objectives, and the detailed explanation of how the method operates. The extensive empirical studies conducted by the authors provide strong evidence of the attack's effectiveness, enhancing the paper's overall quality. Additionally, the paper goes beyond the attack itself by including basic experiments for defense strategies, demonstrating a comprehensive approach to addressing the issue.

However, some concerns were raised by reviewers. One reviewer pointed out similarities between the attack method and the existing HotFlip technique, seeking clarification regarding the technical contributions of the paper. Additionally, the paper lacks comparisons to state-of-the-art (STOA) works in the field. The paper's focus on a specific type of dense retrieval architecture, the bi-encoder model, should be made clear in the abstract and introduction, and the paper should acknowledge other relevant architectures and cite appropriate literature. Lastly, the evaluation on the BEIR dataset should consider the similarity of queries in different collections relative to the training data to provide a more accurate assessment of poisoning vulnerability.